

# Plating versus intramedullary fixation for mid-shaft clavicle fractures: a systemic review and meta-analysis

Yan Gao[1], Wei Chen[2,3], Yue-Jv Liu[2,3], Xu Li[2,3], Hai-Li Wang[2,3] and Zhao-yu Chen[2,3,4]

[1] Department of Endocrinology, The Second Hospital of Hebei Medical University, Shijiazhuang, Hebei Province, China
[2] Department of Orthopedics, The Third Hospital of Hebei Medical University, Shijiazhuang, Hebei Province, China
[3] Key Laboratory of Orthopedic Biomechanics of Hebei Province, The Third Hospital of Hebei Medical University, Shijiazhuang, Hebei Province, China
[4] Arthritis Clinic and Research Center, Peking University People's Hospital, Beijing, China

Corresponding author
Zhao-yu Chen, hbchen-zhaoyu@163.com

## ABSTRACT

**Background.** Plate fixation and intramedullary fixation are the most commonly used surgical treatment options for mid-shaft clavicle fractures; the latter method has demonstrated better performance in some studies.

**Objectives.** Our aim was to critically review and summarize the literature comparing the outcomes of mid-shaft clavicle fracture treatment with plate fixation or intramedullary fixation to identify the better approach.

**Search Methods.** Potential academic articles were identified from the Cochrane Library, MEDLINE (1966-2015.5), PubMed (1966-2015.5), EMBASE (1980-2015.5) and ScienceDirect (1966-2015.5). Gray studies were identified from the references of the included literature.

**Selection Criteria.** Randomized controlled trials (RCTs) and non-RCTs comparing plate fixation and intramedullary fixation for mid-shaft clavicle fracture were included.

**Data Collection and Analysis.** Two reviewers performed independent data abstraction. The $I^2$ statistic was used to assess heterogeneity. A fixed- or random-effects model was used for the meta-analysis.

**Results.** Six RCTs and nine non-RCTs were retrieved, including 513 patients in the intramedullary fixation group and 521 patients in the plating group. No significant differences in terms of the union rate and shoulder function were found between the groups. Patients in the intramedullary fixation group had a shorter operative time, less blood loss, smaller wound size, and shorter union time than those in the plating group. With respect to complications, significant differences were identified for all complications and major complications (wound infection, nonunion, implant failures, transient brachial plexopathy, and pain after 6 months). Similar secondary complications (symptomatic hardware, hardware irritation, prominence, numbness, hypertrophic callus) were observed in both groups.

**Conclusions.** Intramedullary fixation may be superior to plate fixation in the treatment of mid-shaft clavicle fractures, with similar performance in terms of the union rate and shoulder function, better operative parameters and fewer complications.

## INTRODUCTION

Clavicular fractures are common, comprising 2.6–10% of all fractures (*O'Neill et al., 2011*), and approximately 80% of clavicle fractures involve the middle shaft. The majority of these clavicle fractures occur in patients who are younger than 40 or older than 70 years (*Kim & McKee, 2008*). Non-operative therapy can be successful, with high union rates that may be maintained for decades. Although only 7% of patients with clavicle fractures developed nonunion after conservative treatment, 46% of patients in the study of *Nowak, Holgersson & Larsson (2004)* had persistent symptoms 10 years after the injury. Based on recent studies, there has been a trend toward the surgical treatment of clavicle fractures.

More than 50% of clavicle fractures are displaced (*Postacchini et al., 2002*). Displaced fractures carry a risk of malunion or nonunion, both of which result in non-satisfactory function. According to *Zlowodzki et al. (2005)*, the overall nonunion rate for clavicle fractures was 5.9%, whereas the rate for displaced fractures was 15.1%. Recent prospective randomized studies have reported superior functional results with intramedullary nailing (*Smekal et al., 2009*) or plating (*Society, 2007*) compared with conservative treatment. Moreover, a recent meta-analysis revealed a significantly lower nonunion rate after surgical treatment (*Zlowodzki et al., 2005*). Early surgical intervention has therefore been suggested to improve outcomes and to decrease the rates of nonunion and symptomatic malunion in mid-shaft clavicle fractures with the following features: open injury, shortening or displacement >20 mm, multiple trauma, floating shoulder, or cosmetic concerns (*Society, 2007*; *Hill, McGuire & Crosby, 1997*).

Plate fixation is the standard surgical therapy for mid-shaft clavicular fractures (*Ali & Lucas, 1978*). A 2.2% nonunion rate was reported in a review synthesizing the results of earlier studies on displaced clavicular fractures treated by plate fixation (*Zlowodzki et al., 2005*). However, clavicular plates require larger skin incisions and extensive soft tissue stripping, which increase the risk for nonunion and wound infection. Moreover, clavicle re-fracture occurred after plate removal in 0–8% of the patients (*Bostman, Manninen & Pihlajamaki, 1997*; *Poigenfurst, Rappold & Fischer, 1992*). The Knowles pin, the Rockwood pin, and the titanium elastic nail (TEN) have been developed to minimize postoperative complications (*Jubel et al., 2003*). From a biomechanical perspective, intramedullary implant positioning is ideal (*Mueller et al., 2008*). With the advantages of intact hematoma maintenance, less soft tissue dissection and periosteal stripping, all of which can accelerate fracture healing, intramedullary fixation has been gaining attention for its superior performance. Unfortunately, hardware migration (including medial migration and lateral perforation) has been a problem with intramedullary fixation. The rate of TEN migration ranges between 4.5% and 26.6% in the literature (*Meier, Grueninger & Platz, 2006*; *Kettler et al., 2007*). Overall, different complication rates were reported for these two fixation methods, but no significant differences were

noted for most of them. Significantly more instances of symptomatic hardware, infection, nonunion, wound dehiscence, and refractures were reported with plate fixation than with intramedullary fixation in studies (*Lee et al., 2007*; *Saha et al., 2014*; *Narsaria et al., 2014*). Furthermore, mobilization after intramedullary fixation requires more attention because conduction stabilization is weaker with intramedullary fixation than with plate fixation (*Frigg et al., 2009*).

Several meta-analyses comparing plate and intramedullary fixation were published from 2011 to 2015 (*Duan et al., 2011*; *Houwert et al., 2012*; *Barlow, Beazley & Barlow, 2013*; *Zhu et al., 2015*). However, the studies included were not exactly comparable, and fewer than 5 RCTs or quasi-RCT (qRCT) studies were included. Here, we review and summarize all of the RCTs and non-RCTs comparing plate and intramedullary fixation in the hopes of presenting useful data to identify the better treatment choice and to confirm the findings of these studies.

## MATERIAL AND METHODS

### Inclusion and exclusion criteria

Trials with the following characteristics were included: (1) RCTs or non-RCTs, (2) patients with midshaft clavicle fractures from trauma and without pathological fractures, (3) comparison of the results of intramedullary and plating fixation, and (4) full-text articles. We excluded articles that were duplicate reports of earlier trials or post-hoc analyses of RCT data and articles without an available full-text version. Studies including patients suffering multiple traumas were also excluded.

### Search strategy

Electronic searches of the Cochrane Library, MEDLINE (1966-2015.5), PubMed (1966-2015.5), EMBASE (1980-2015.5) and ScienceDirect (1966-2015.5) as well as other Internet databases were performed to identify trials, according to the Cochrane Collaboration guidelines. We used the following search terms and different combinations of Medical Subject Heading (MeSH) terms and textual words: ''clavicle or clavicular,'' ''fracture,'' ''midshaft or mid-shaft,'' ''intramedullary,'' ''plate, plates or plating.'' Manual searches, including those of the reference lists of all included studies, were used to identify trials that the electronic search may have failed to identify. Two reviewers (Yan Gao and Zhao-Yu Chen) independently assessed the titles and abstracts of all of the reports identified by the electronic and manual searches. There was no restriction on language. When inclusion was unclear based on the abstracts alone, the full-text articles were retrieved. Any disagreements were resolved through discussion.

### Assessment of methodological quality

Quality assessment of RCTs and non-RCTs was conducted according to a modification of the generic evaluation tool used by the Cochrane Bone, Joint and Muscle Trauma Group (*Handoll et al., 2008*) or the index for non-randomized studies form (*Slim et al., 2003*). The methodological quality of each trial was scored from 0 to 24. Disagreements were resolved by consensus or by consultation with the senior reviewer (Wei Chen).

## Data extraction

Two authors (Yan Gao and Yue-Jv Liu) independently extracted data from the included articles. Information regarding the study design, patient demographics, inclusion and exclusion criteria, interventions, outcomes, follow-up duration and rate of loss to follow-up for each treatment group were extracted. When continuous outcomes were published as the median and range in the original papers, the mean value and standard deviation were estimated using the formula provided by *Hozo, Djulbegovic & Hozo (2005)* Data were managed using Review Manager (RevMan) 5.1 software (The Nordic Cochrane Centre, The Cochrane Collaboration, Copenhagen, Denmark). We attempted to contact authors for supplementary information when the reported data were inadequate.

## Data analysis and statistical methods

The meta-analysis was conducted using RevMan 5.1 for Windows (Cochrane Collaboration, Oxford, United Kingdom). Statistical heterogeneity was assessed for each study, using a standard Chi square test, with significance set at a $P$ value of 0.1, which was measured by the $I^2$ statistic. When $I^2 > 50\%$, $P < 0.1$ was considered to be significant heterogeneity (*Higgins et al., 2003*). Therefore, a random-effects model was applied for data analysis (*Lau, Ioannidis & Schmid, 1997*). A fixed-effects model was used when no significant heterogeneity was found. In cases of significant heterogeneity, subgroup analysis was performed to investigate sources. The odds ratio (OR) and 95% confidence interval (CI) were calculated for dichotomous outcomes, whereas the mean difference (MD) and 95% CI were used for continuous outcomes.

# RESULTS

## Literature search

Figure 1 shows a flow chart of the study selection and inclusion process. The search strategy identified 194 citations; of these, six RCTs and nine non-RCTs met the predefined inclusion criteria for data extraction and meta-analysis.

## Study characteristics

Individual patient data were obtained from these articles. Population information is summarized in Table 1. These studies included 513 patients in the intramedullary fixation (IF) group and 521 patients in the plate fixation (PF) group, excluding those lost to follow up. Between-group differences in the baseline characteristics were not found. The quality assessment scores of the studies ranged from 17 to 20.

## Risk of bias assessment

For the RCTs, unclear blindness was the major problem (details are provided in Table 2). For the nine non-RCTs, no prospective calculation of the sample size was described. Moreover, no information regarding the unbiased assessment of study endpoints was available. Only five studies reported the relevant information regarding the prospective collection of data. The methodological quality assessment is illustrated in Fig. 2.

**Table 1  Characteristics of included studies.**

| Study | Time | Type | Invention | | Age(years) | | Gender(F/M) | | Follow-up(months) | |
|---|---|---|---|---|---|---|---|---|---|---|
| | | | IF | PF | IF | PF | IF | PF | IF | PF |
| Lee YS et al. | 2007 | RCT | Knowles pin | DCP | 60.4(50–81) | 56.7(52–79) | 32(19/13) | 30(17/13) | 30 | |
| Lee YS et al. | 2008 | RCT | Knowles pin | DCP, tubular and reconstruction plate | 40.1 | 38.2 | 56(19/37) | 32(12/20) | 12 | |
| Ferran NA et al. | 2010 | RCT | Rockwood Pin | LC-DCP | 23.8(13–42) | 35.4(16–53) | 17(14/3) | 15(13/2) | $12.7 \pm 3.5$ | $12.1 \pm 5.7$ |
| Assobhi JE et al. | 2011 | RCT | Titanium elastic nail | 3.5 mm reconstruction plate | $30.3 \pm 4.8$ | $32.6 \pm 5.9$ | 19(16/3) | 19(17/2) | $14.5 \pm 1.5$ | $18.6 \pm 3.8$ |
| Narsaria N et al. | 2014 | RCT | Titanium elastic nail | 3.5 mm DCP | $38.9 \pm 9.1$ | $40.3 \pm 11.2$ | 33(9/24) | 32(6/26) | 24 | |
| Saha P et al. | 2014 | RCT | Titanium elastic nail | Locking plate | $33.3 \pm 11.8$ | $33.0 \pm 12.6$ | 34(4/30) | 37(7/30) | $24.6 \pm 2.4$ | $25.1 \pm 3.3$ |
| S, Thyagarajan D et al. | 2009 | nRCT | Rockwood Pin | LC-DCP | 28(15–56) | 32.1(17–46) | 17(1/16) | 17(2/15) | 5.9(4–11) | |
| Liu HH et al. | 2010 | nRCT | Titanium elastic nail | Reconstruction LCP | $33.6 \pm 13.5$ | $31.7 \pm 9.7$ | 51(19/32) | 59(30/29) | 17.7(12–27) | |
| Kleweno CP et al. | 2011 | nRCT | Rockwood Pin | Reconstruction plate or locking plate | 35(16–56) | 28(16–46) | 18(3/15) | 14(4/10) | 8(3–28) | 17(4–58) |
| Fu TH et al. | 2012 | nRCT | Knowles pin | Reconstruction plate | $35.2 \pm 14.5$ | $39.9 \pm 14.8$ | 53(15/38) | 40(17/33) | 15(12–153) | 14(12–92) |
| Chen YF et al. | 2012 | nRCT | Titanium elastic nail | 3.5 mm reconstruction plates | 38(26.5–58) | 46.5(36.5–58.8) | 25(15/10) | 32(14/18) | 12 | |
| Tarng YW et al. | 2012 | nRCT | Titanium elastic nail | Reconstruction plates | 34.3(20–59) | 36.5(19–63) | 57(16/41) | 84(23/61) | 24 | |
| Wijdicks FJ et al. | 2012 | nRCT | Titanium elastic nail | Reconstruction plate or locking plate | $39.4 \pm 14.1$ | $33.1 \pm 15.6$ | 43(10/33) | 47(14/33) | 6(5–12) | 8(2–15) |
| Wenninger JJ et al. | 2013 | nRCT | Rigid Hagie pin | 3.5 mm reconstruction plate or LC-DCP | 25.2(18–51) | 26.9(20–49) | 33(1/32) | 29(3/26) | 12 | |
| Jones LD et al. | 2014 | nRCT | Titanium elastic nail | N | N | N | 25 | 24 | 30(12–54) | |
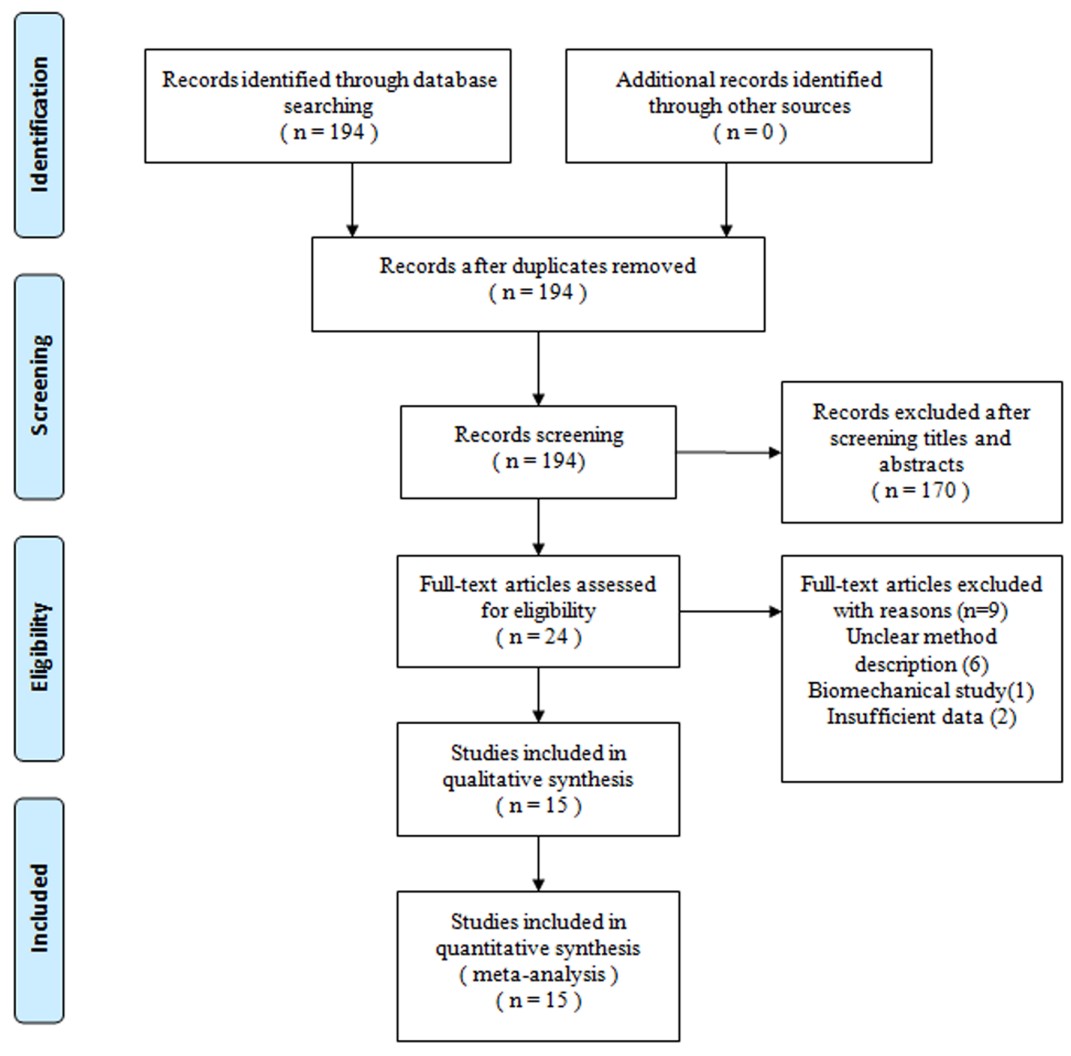

**Figure 1  Flow chart showing identification and selection of cases.**

## Meta-analysis outcomes

### Blood loss

Blood loss was reported in five studies (*Liu et al., 2010b*; *Chen et al., 2012*; *Tarng et al., 2012*; *Saha et al., 2014*; *Narsaria et al., 2014*), including 200 patients in the IF group and 244 patients in the PF group. The mean blood loss and standard deviation were estimated from the study of *Tarng et al. (2012)*. Greater blood loss was observed in the PF group than in the IF group (MD = −64.14; 95% CI: [−66.88 to −61.40]; $P < 0.001$; Fig. 3).

### Operative time

Operative time was reported in seven studies (*Lee et al., 2008*; *Liu et al., 2010b*; *Assobhi, 2011*; *Chen et al., 2012*; *Tarng et al., 2012*; *Narsaria et al., 2014*; *Saha et al., 2014*), including 275 patients in the IF group and 295 patients in the PF group. The mean operative time and standard deviation were estimated from the studies of *Tarng et al. (2012)* and

**Table 2  Quality assessment for randomized trials.**

| Quality assessment for randomized trials | Lee YS (2007) | Lee YS | Ferran NA | Assobhi JE | Narsaria N | Saha P |
|---|---|---|---|---|---|---|
| Was the assigned treatment adequately concealed prior to allocation? | 1 | 1 | 2 | 2 | 2 | 1 |
| Were the outcomes of participants who withdrew described and included in the analysis? | 2 | 2 | 2 | 2 | 2 | 2 |
| Were the treatment and control group comparable at entry? | 2 | 2 | 2 | 2 | 2 | 2 |
| Were the outcome assessors blinded to treatment status? | 2 | 0 | 2 | 0 | 0 | 0 |
| Were the participants blind to assignment status after allocation? | 0 | 0 | 0 | 0 | 0 | 0 |
| Were the treatment providers blind to assignment status? | 0 | 0 | 0 | 0 | 0 | 0 |
| Were care programs, other than the trial options, identical? | 2 | 2 | 2 | 2 | 2 | 2 |
| Were the inclusion and exclusion criteria clearly defined? | 2 | 2 | 2 | 2 | 2 | 2 |
| Were the interventions clearly defined? | 2 | 2 | 2 | 2 | 2 | 2 |
| Were the outcome measures used clearly defined? | 2 | 2 | 2 | 2 | 2 | 2 |
| Were diagnostic tests used in outcome assessment clinically useful? | 2 | 2 | 2 | 2 | 2 | 2 |
| Was the surveillance active, and of clinically appropriate duration? | 2 | 2 | 2 | 2 | 2 | 2 |

*Narsaria et al. (2014)*. The operative time was shorter in the IF group than in the PF group (MD $= -22.30$; 95% CI [$-30.42$ to $-14.18$]; $P < 0.001$; Fig. 4). High heterogeneity ($I^2 = 95\%$) was found and was not significantly reduced when the analysis was performed using only four RCTs (*Lee et al., 2008*; *Assobhi, 2011*; *Narsaria et al., 2014*; *Saha et al., 2014*) (MD $= -24.72$; 95% CI [$-34.40$ to $-15.04$]; $P < 0.001$).

### Wound size

Six studies reported the size of the surgical wound (*Lee et al., 2007*; *Liu et al., 2010b*; *Assobhi, 2011*; *Tarng et al., 2012*; *Fu et al., 2012*; *Narsaria et al., 2014*). The wound size was smaller in the IF group than in the PF group (MD $= -5.07$; 95% CI [$-6.00$ to $-4.13$]; $P < 0.001$; $I^2 = 97\%$, Fig. 5). This result did not significantly change when the result of *Liu et al. (2010a)* was excluded because of a significantly larger wound size (MD $= -4.36$; 95% CI [$-5.22$ to $-3.50$]; $P < 0.001$).

### Hospital stay

The length of hospital stay was reported in five studies, including 3 RCTs (*Lee et al., 2007*; *Assobhi, 2011*; *Narsaria et al., 2014*) and 2 non-RCTs (*Liu et al., 2010b*; *Tarng et al., 2012*). Two studies (*Liu et al., 2010b*; *Assobhi, 2011*) provided the original mean and standard deviation, while the other studies published the median and range. The length of hospital stay was shorter in the IF group than in the PF group, regardless of whether the meta-analysis was performed on all of the included studies (MD $= -1.31$; 95% CI [$-1.69$ to $-0.93$]; $P < 0.001$; $I^2 = 80\%$; Fig. 6), on RCTs only (MD $= -1.57$; 95% CI [$-2.30$ to $-0.84$]; $P < 0.001$; $I^2 = 85\%$), or on only the studies providing the original mean and standard deviation (MD $= -0.98$; 95% CI [$-1.36$ to $-0.59$]; $P < 0.001$).
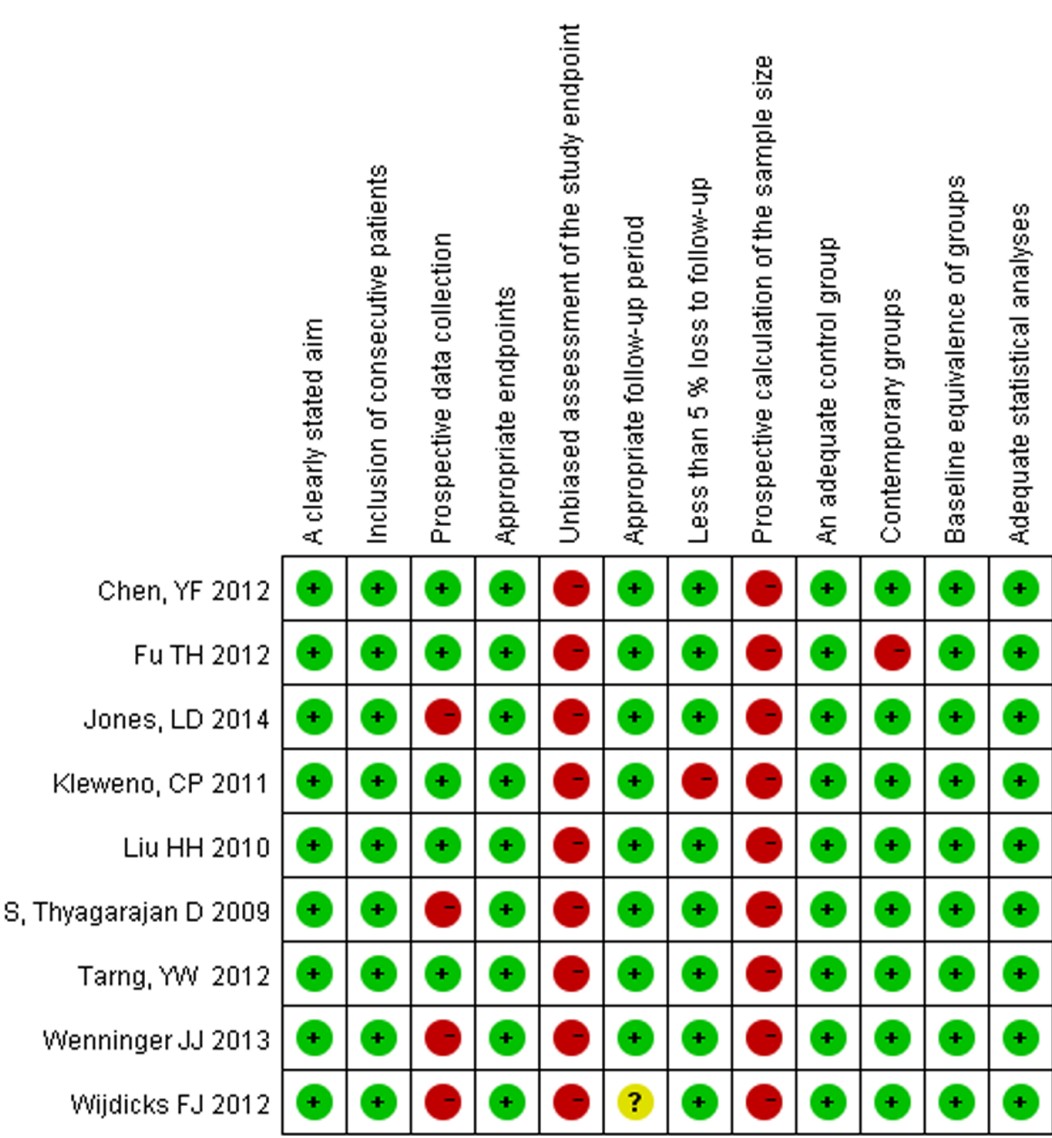

**Figure 2** Quality assessment for non-randomized trials.

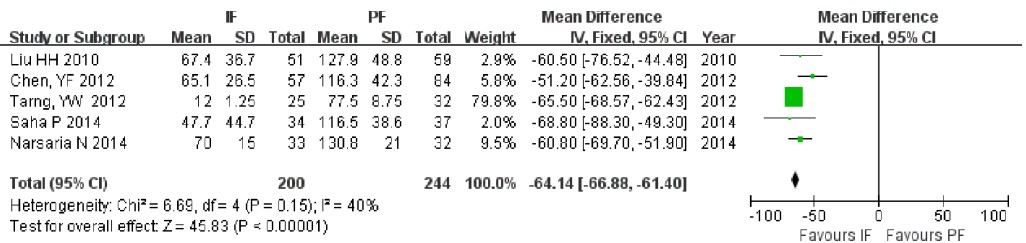

**Figure 3** Forest plot showing blood loss in the two groups.

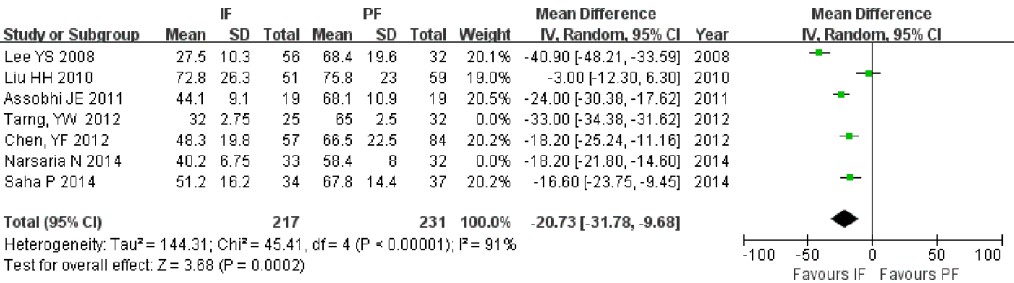

**Figure 4** Forest plot showing operative time in the two groups.

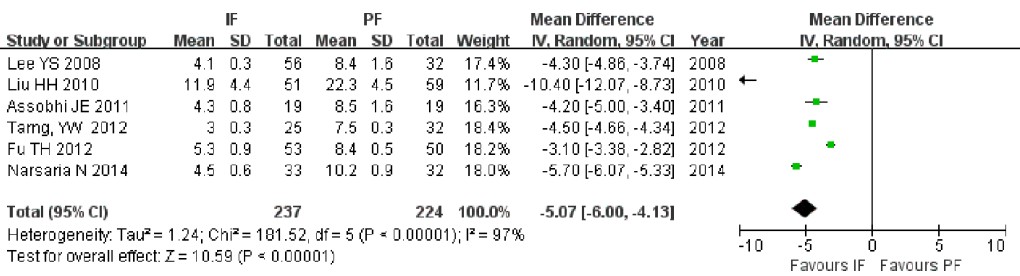

**Figure 5** Forest plot showing wound size in the two groups.

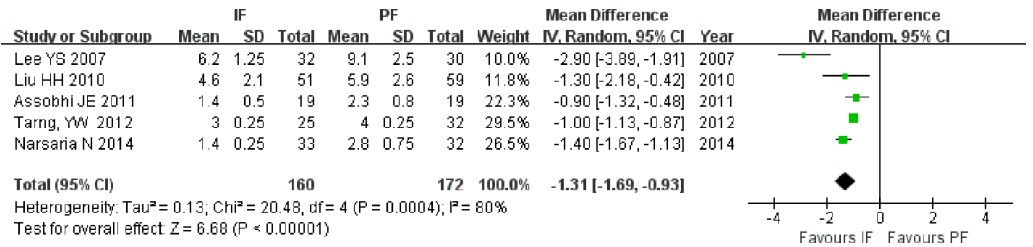

**Figure 6** Forest plot showing hospital stays in the two groups.

### Union rate and union time

The union rate was reported in all of the included studies. Neither total nor subgroup analysis of RCTs and non-RCTs revealed significant differences between the fixation methods (OR = 1.41; 95% CI [0.73–2.75]; $P = 0.31$), (OR = 2.20; 95% CI [0.57–7.77]; $P = 0.27$), and (OR = 1.23; 95% CI [0.56–2.66]; $P = 0.61$; Fig. 7). Union time data were extracted from four RCTs (*Lee et al., 2008*; *Assobhi, 2011*; *Saha et al., 2014*; *Narsaria et al., 2014*) and two non-RCTs (*Liu et al., 2010b*; *Chen et al., 2012*). The results based on all of the included studies showed shorter union times in the IF group than in the PF group (MD = −16.25; 95% CI [−28.03 to −4.47]; $P = 0.007$; Fig. 8). The union time was also shorter in the IF group than in the PF group when the analysis was performed only on RCTs (MD = −26.40; 95% CI [−46.20 to −6.61]; $P = 0.009$).

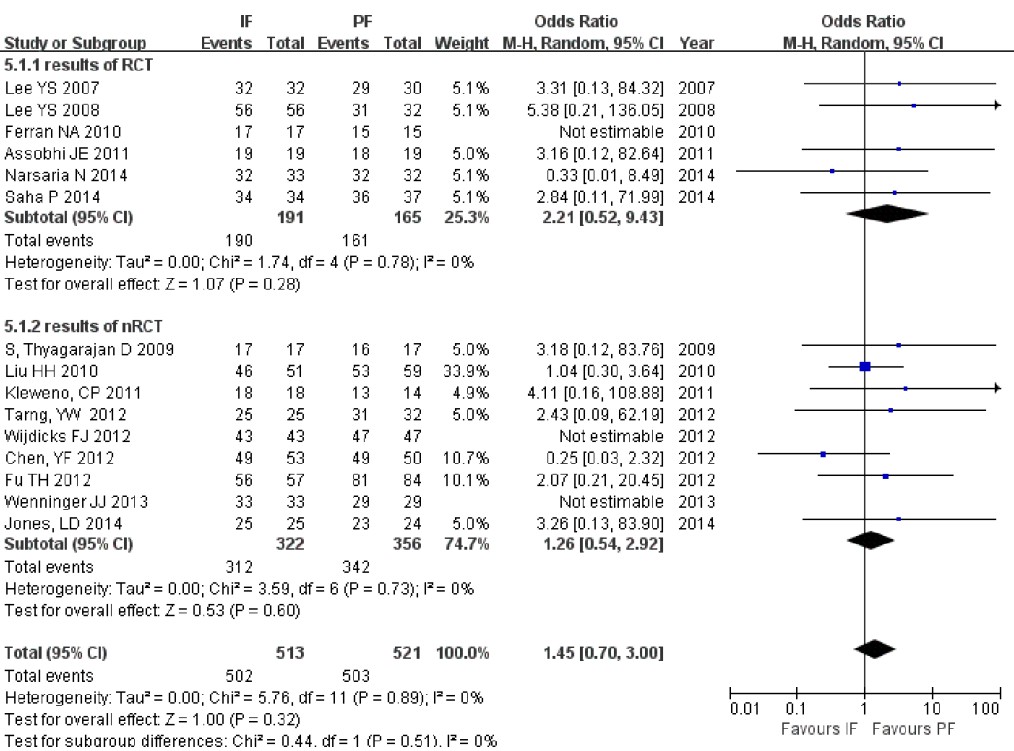

**Figure 7** Forest plot showing union rate in the two groups.

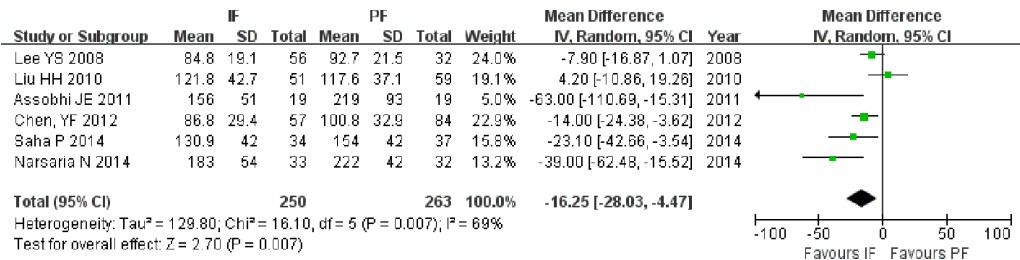

**Figure 8** Forest plot showing union time in the two groups.

## *Shoulder score*

Six RCTs (*Lee et al., 2007*; *Lee et al., 2008*; *Ferran et al., 2010*; *Assobhi, 2011*; *Saha et al., 2014*; *Narsaria et al., 2014*) and three non-RCTs (*S et al., 2009*; *Liu et al., 2010b*; *Tarng et al., 2012*), including 260 patients in the IF group and 273 patients in the PF group, published the Shoulder score. The meta-analysis based on all included studies did not show superior function in either group (MD = 1.82; 95% CI [−0.05–3.70]; $P = 0.06$; Fig. 9); similar results were observed when the analysis included only RCTs (MD = 1.42; 95% CI [−0.68–3.52]; $P = 0.19$).

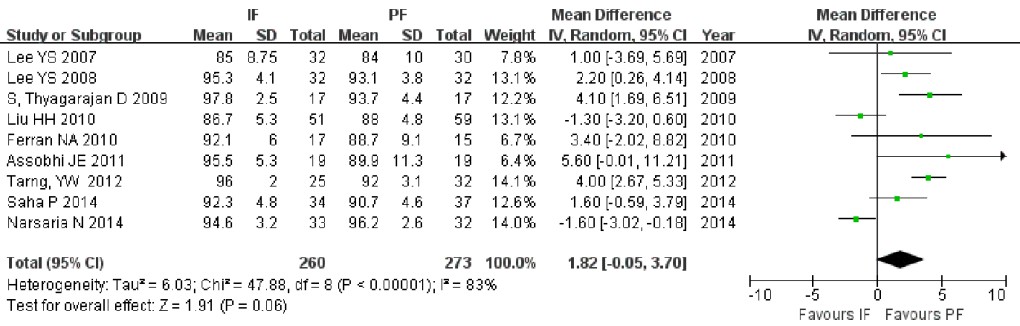

**Figure 9** Forest plot showing shoulder score in the two groups.

### Complications

All included studies reported on complications related to IF or PF, and more complications occurred in the PF group than in the IF group (OR = 0.43; 95% CI [0.25–0.76]; $P = 0.003$; Fig. 10). When only the major complications (wound infection, nonunion, implant failures, transient brachial plexopathy, and pain after 6 months) were considered, the subgroup analysis also showed more major complications in the PF group than in the IF group (OR = 0.52; 95% CI [0.33–0.81]; $P = 0.004$). No significant differences were found in secondary complications (OR = 0.43; 95% CI [0.16–1.15]; $P = 0.09$).

## DISCUSSION

With lower nonunion and malunion rates and better function, especially for displaced fractures, early operative interventions have become the preferred treatment method for mid-shaft clavicular fractures. This meta-analysis reviewed and summarized data from the literature comparing plating fixation and intramedullary fixation. Both surgical techniques showed similar performance in terms of the union rate and shoulder function. With less blood loss, shorter operative time, shorter hospital stay, shorter time to union, and fewer major complications, intramedullary fixation showed better results than plate fixation. Similar results were reported in a meta-analysis of open reduction and internal fixation versus TEN by *Duan et al. (2011)*, who observed no significant difference in treatment effects but more side effects in the patients treated with plates (*Houwert et al., 2012*).

The power of a meta-analysis depends on the quality of the included studies. To provide better evidence for clinical application, we searched and included RCTs and non-RCTs with quality assessment scores ranging from 17 to 20. For the RCTs, the patients were randomized into two groups using an envelope method in three studies and an alternating one-by-one allocation method in the other studies. The major problem was that not enough information was provided to indicate that participants and treatment providers were blinded to the assignment status. Only two studies described assessor blinding. No prospective calculation of the sample size was described in the non-RCTs. Moreover, the assessment of the study endpoints was biased. All of these shortcomings weaken the level of evidence.

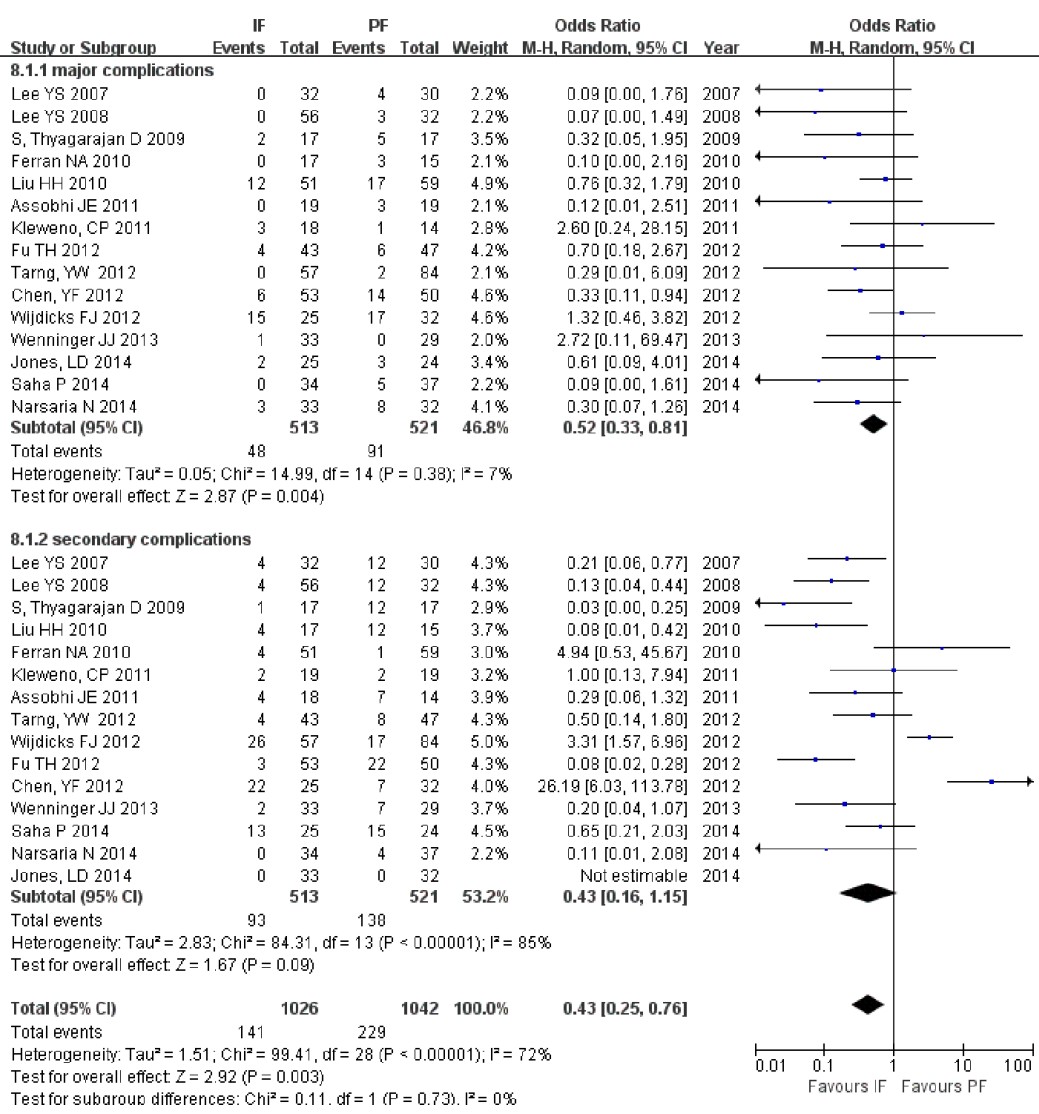

**Figure 10 Forest plot showing complications in the two groups.**

Compared with open reduction and internal fixation with plates, intramedullary fixation with nails or pins has minimally invasive characteristics, including smaller skin incisions and reduced soft tissue stripping, which are attractive qualities. Therefore, less blood loss, a shorter operative time and a shorter hospital stay definitely benefitted the patients who received intramedullary fixation. Preservation of the soft tissue envelope and periosteum increases the chances of healing (*Liu et al., 2010b*). Stable fixation is another basic principle that can ensure fracture union.

The S-shaped clavicle, as the only bone connecting the upper limb with the body, behaves in a complex manner. Plate fixation is better able to resist the bending and torsional forces exerted during elevation of the upper extremity above shoulder level, thereby potentially providing a stronger construction for early rehabilitation protocols (*Golish et al., 2008*). Therefore, similar union rates can be realized by either intramedullary fixation

or plate fixation because of the sufficient blood supply and stable fixation, as observed in our present review.

Union is the fundamental factor for shoulder function recovery after fractures; for this reason, no significant difference was observed between the union rates of the groups. However, union is not the only factor that can influence clinical function after a clavicular fracture (*Lazarides & Zafiropoulos, 2006*). The clavicle length plays an important role in maintaining anatomical relationships (*McKee et al., 2006*). Malunion with more than 15 mm of shortening has been reported to result in weakness of the glenohumeral extension and abduction (*Ledger et al., 2005*). Compared with fixation using intramedullary devices, fixation with a contoured plate is the best choice for maintaining the S-shape and length of the clavicle, especially for comminuted fractures. TEN, a newly designed intramedullary device with a curved tip, is flexible and fixed in the cancellous substance of the distal clavicle, and it can better accommodate the S-shaped contour of the clavicle and more tightly adhere to the cortex compared with K-wires, screws, or pins. Surprisingly, clavicle shortening was reported by *Chen et al. (2012)*, with 2 cases in the TEN group, and by Wijdicks et al., with 6 cases in the elastic stable intramedullary nail group; additionally, Saha et al., reported clavicle shortening by $6.29 \pm 3.75$ mm in the TEN group. We suggest that both plate and intramedullary devices can restore the clavicle to an ideal length to regain most shoulder function. The residual shortening after plate or intramedullary fixation does not significantly influence shoulder function. However, many other factors may also contribute to shoulder function.

As in the study of *Assobhi (2011)*, complications were divided into major complications (including nonunion, infection, implant failure, re-fracture, and transient brachial plexopathy) and secondary complications regarding cosmesis. Extensive soft tissue exposure carries significant risks for nonunion and wound infection in plate fixation. More re-fractures occurred in the plating group because rigid fixation could introduce stress shielding, resulting in bone weakness. Moreover, screw holes may act as focal points for stress, thereby leading to re-fracture. Hardware prominence is the major problem for intramedullary devices. In the literature, the reported rate of hardware prominence ranged from 5.2% to 38.8% with the use of an elastic intramedullary nail (*Smekal et al., 2009*; *Jubel et al., 2003*; *Frigg et al., 2009*). Careful surgical manipulation is necessary to avoid prominence. Additionally, cosmetic problems, including hypertrophic scarring and implant prominence, were reported in some studies that showed better results for pin fixation; the lack of such cosmetic problems could be an important factor contributing to the acceptance of this method.

The physiotherapy mentioned in some studies included sling protection for 2–4 weeks post-operatively, instructions for gentle and passive range of motion shoulder movements, normal daily activities after a 4-week postoperative period, or weight lifting and the return to full activities after complete fracture healing. Because the activities were similar for both procedures, we trusted that no significant differences exist between the two procedures based on the information we could obtain.

Some limitations should be considered when interpreting these analyses. First, various intramedullary devices and plates were applied in the included studies. Accordingly,

we could only report the average performance; thus, more attention must be focused on choosing a specific pin or nail in clinical practice. The TEN was the most frequently used intramedullary device; more analysis regarding this implant will be necessary in the future. Moreover, classifying some complications as minor may be inappropriate because cosmetic problems are also important issues for female patients. More details of special interest should be considered in the future. As we were limited by unavailable data on the facilities in different cities, we could not analyze the impact of these different facilities on decision making and outcomes. Classifying and comparing outcomes in a chronological order is also difficult to perform. Despite these limitations, we have provided the most comprehensive data comparing plate fixation with intramedullary fixation in the treatment of clavicle fractures.

## CONCLUSION

Based on the results of our meta-analysis, both plate and intramedullary fixation can achieve similar union rates and shoulder function. Considering the better performance of intramedullary fixation in terms of operative parameters and complications, we recommend the application of intramedullary fixation for displaced mid-shift clavicular fractures. More studies focused on comminuted fractures will be necessary in the future.

## ACKNOWLEDGEMENTS

The authors thank Dr. Zhi-jun Li (Tianjin Medical University General Hospital) and Professor Feng-shi Ma (Department of Mathematics, Tianjin University) for their helping with statistical section of the paper.

### Funding

The authors received no funding for this work.

### Competing Interests

The authors declare there are no competing interests.

### Author Contributions

- Yan Gao performed the experiments, analyzed the data, contributed reagents/materials/analysis tools, wrote the paper, prepared figures and/or tables, reviewed drafts of the paper.
- Wei Chen conceived and designed the experiments, performed the experiments.
- Yue-Jv Liu analyzed the data, contributed reagents/materials/analysis tools.
- Xu Li contributed reagents/materials/analysis tools.
- Hai-Li Wang prepared figures and/or tables, reviewed drafts of the paper.
- Zhao-yu Chen conceived and designed the experiments, wrote the paper, reviewed drafts of the paper.

## Data Availability

Raw data can be found in Data S1.

## Supplemental Information

Supplemental information for this article can be found online at http://dx.doi.org/10.7717/peerj.1540#supplemental-information.

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
