# Peer review of "Plating versus intramedullary fixation for mid-shaft clavicle fractures: a systemic review and meta-analysis"

_PeerJ, doi:10.7717/peerj.1540_

## Round 0.1 · original submission · Major Revisions

Please thoroughly address the reviewers', particular the 2nd reviewer's comments with point-by-point explanation. Thank you.

Reviewer 1 ·

Basic reporting

There have been a few mata-analysis articles in prior literature regarding the comparison of clinical outcomes between plate and intramedullary fixation for mid-shaft clavicle fractures, in which Zhang, et al (Zhang et al. Scand J Trauma Resusc Emerg Med. 2015 Mar 20;23:27) and Zhu, et al (Zhu, et al. Int Orthop. 2015 Feb;39(2):319-28. ) drew almost the same conclusion as this paper, so the introduction and background to the work of this paper should be described more detailedly to reveal its feature.

Experimental design

No comments.

Validity of the findings

Controversies exist regarding the optimal option between plate or intramedullary fixation for treatment of mid-shaft clavicle fracture partly because there are different types of the fracture, and their optimal fixation methods may be different, moreover, different kinds of plate or intramedullary device should have different fixation effects even applied for a same type of this fracture. So, strictly speaking, there is no universal optimal fixation method suiting for all types of mid-shaft clavicle fracture. This authors of this meta-analysis paper drew a conclusion that intramedullary fixation may be superior to plate fixation in the treatment of mid-shaft clavicle fractures, with similar performance in union rate and shoulder function, better operative parameters and fewer complications. But there are also limitations as the authors mentioned, various intramedullary devices and plates were applied in the included studies, and no types of fracture were listed for comparison in these studies.

Additional comments

As mentioned above,
1) The introduction and background to the work of this paper should be described more detailedly to reveal its feature; and
2) if possible, a meta-analysis with more detailed fracture types and kinds of plate or intramedullary device may provide more reasonable evident for choice of fixation methods for mid-shaft clavicle fracture.

Reviewer 2 ·

Basic reporting

No Comments

Experimental design

No Comments

Validity of the findings

No comments

Additional comments

In this review, authors compare plating versus intramedullary fixation for mid-shaft clavicle fractures which is one of the most common fractures in orthopedic practice. The study has compared the different outcomes of the two procedures using systemic review and meta-analysis.


Major Issues:
• The study in general lacks demographic information of the cases obtained. As authors chose to compare the two methods, they focused on various clinical outcomes of these methods ignoring the differences in demographic information of the sample selected.
Demographic information such as gender, age, and location of the operation should have been compared and stratified according to different outcomes (Primary and secondary) as they may change the results of the study, provide us with better understanding of the outcome, help achieving appropriate decision making of which method to choose according to each patients individual data.
• The authors mentioned difficulties associated with measuring and assessing different techniques and equipment used in both procedures. As participated studies in this review have been selected in time period between 1966-2015. Classifying and comparing outcomes in a chronological order would have minimized the difference questioned regarding this variable. Keeping in mind that clinical outcomes would also change as a natural result of the recent development in medical and surgical care in general.
• The study did not mention long term outcomes of surrounding joints of the fracture, mainly shoulder joint and sternoclavicular joint.
• The facility where the procedure took place was not mentioned too. This would have great impact on the decision made at the time of the surgery as well as the outcomes. For example, some hospitals serving a high socioeconomic areas would prefer procedures with less cosmetic complications and would be at the same time equipped with better staff and facilities.
• There was no mention of physiotherapy in both procedures. How would physiotherapy affect the outcome of this fracture? And would it be affected differently between the two methods.


Minor Issues:
• In line 78-79, it would have been better to mention which method had fewer complications.
• In line 124, it is mentioned that “The concept of early mobilization after intramedullary implantation needs more attention”. What do you “The author” means by more attention? Is it better to early mobilize the patient after the procedure or not? Or more data is needed regarding this practice?
• In lines 263 and 318, it would be helpful to mention the secondary complications.

---

## Round 0.2 · Minor Revisions

Before the final acceptance, the manuscript requires extensive professional editing. Please do so and provide a tracked and a clean version and, if you use one, a certificate from an accredited professional English Editing company. Thank you.

---

## Round 0.3 · Minor Revisions

Thank you for your effort in the revision. However, the certificate is outdated because it is marked as May 27th, 2015, prior to the revision. Also, it seems like that the authors have added a few paragraphs in tracked mode, but there are English issues, such as: "contribute welcome", "similar for", and more, which require professional editing. Overall, a certificate dated after 10/15/2015 from professional editing services such as AJE is required before its final acceptance. Please note, all the newly added paragraphs must be clearly mentioned including line numbers and page numbers which need to be addressed in the rebuttal letter and will be peer-reviewed. Thank you.

---

## Round 0.4 · Minor Revisions

Thank you for the new certificate and tracked files. I applaud you for your effort in a timely response.

However, unfortunately, both reviewers and PeerJ journal stuff have identified an important issue that must be addressed regarding ensuring an appropriate statistical analysis. Accordingly, please 1) carefully revise the statistical section of the paper with input from an appropriate statistician; 2) please provide all the excel sheets containing statistical power result, data analysis as well as raw data with outliers if any as supplemental files for us to review; 3) a rebuttal letter is needed to explain what you have done with the statistical revision.

Please note that although minor revisions were requested to other parts of the paper, the statistical section is a critical step for a paper of this nature, and failure to do so may result in the rejection of the paper. Thank you for your understanding.

---

## Round 0.5 · accepted · Accept

I am pleased to accept this version. Thank you for your support of PeerJ.